# Quantum Phase Transitions in a Generalized Dicke Model

**DOI:** 10.3390/e25111492

**Published:** 2023-10-29

**Authors:** Wen Liu, Liwei Duan

**Affiliations:** 1Key Laboratory of Optical Information Detecting and Display Technology of Zhejiang, Zhejiang Normal University, Jinhua 321004, China; wenliu@zjnu.edu.cn; 2Department of Physics, Zhejiang Normal University, Jinhua 321004, China

**Keywords:** Dicke model, quantum phase transition, mean-field approach, Holstein–Primakoff transformation, quantum fluctuation

## Abstract

We investigate a generalized Dicke model by introducing two interacting spin ensembles coupled with a single-mode bosonic field. Apart from the normal to superradiant phase transition induced by the strong spin–boson coupling, interactions between the two spin ensembles enrich the phase diagram by introducing ferromagnetic, antiferromagnetic and paramagnetic phases. The mean-field approach reveals a phase diagram comprising three phases: paramagnetic–normal phase, ferromagnetic–superradiant phase, and antiferromagnetic–normal phase. Ferromagnetic spin–spin interaction can significantly reduce the required spin–boson coupling strength to observe the superradiant phase, where the macroscopic excitation of the bosonic field occurs. Conversely, antiferromagnetic spin–spin interaction can strongly suppress the superradiant phase. To examine higher-order quantum effects beyond the mean-field contribution, we utilize the Holstein–Primakoff transformation, which converts the generalized Dicke model into three coupled harmonic oscillators in the thermodynamic limit. Near the critical point, we observe the close of the energy gap between the ground and the first excited states, the divergence of entanglement entropy and quantum fluctuation in certain quadrature. These observations further confirm the quantum phase transition and offer additional insights into critical behaviors.

## 1. Introduction

The Dicke model, named after R. H. Dicke, is a prominent example in the field of quantum optics and quantum mechanics [1]. It was initially formulated to describe the collective behavior of a large ensemble of spins (two-level systems or qubits) interacting with a single-mode electromagnetic field within an optical cavity. If only considering a single spin, the Dicke model reduces to the Rabi model, one of the simplest models for studying light–matter interactions [2,3,4]. The Dicke model provides a fundamental framework for exploring collective quantum behavior and the intricate interplay between quantum systems and electromagnetic fields [5,6,7]. Its significance extends across various domains of physics, among which the quantum phase transition in the Dicke model has drawn persistent attention [8,9,10,11,12,13,14].

With a sufficiently large spin–boson coupling strength, the ground state of the Dicke model exhibits a transition from the normal phase to the superradiant phase. This transition is accompanied by macroscopic excitation in the bosonic field [10,11]. The sudden change in the behaviors of the ground state serves as a characteristic signature of the quantum phase transition, which arises from the quantum fluctuation at zero temperature, rather than the thermal fluctuation in the classical phase transition [15]. The quantum phase transition in the Dicke model and its generalizations have been experimentally observed in various platforms, such as Bose–Einstein condensate coupled to an optical cavity [16,17], trapped-ion systems [18], etc. In addition to the superradiant phase transition in the ground state, the Dicke model serves as a versatile prototype to study the excited-state quantum phase transition [19,20,21], nonequilibrium quantum phase transition [22], dynamical phase transition [23], universal dynamics under slow quenches [14], quantum–classical correspondence [24], chaos and thermalization [10,11,25], etc.

Recently, various generalizations of the Dicke model have been proposed, which greatly improve its flexibility. Using squeezed light in the cavity field reduces the necessary spin–boson coupling strength to observe the superradiant phase transition in the Dicke model [26,27,28]. The anisotropic Dicke model introduces different coupling strengths corresponding to the rotating and counter-rotating terms [29,30,31,32]. The two-mode Dicke model yields a richer phase diagram with both first- and second-order quantum phase transitions by introducing an additional bosonic mode [33]. The Rabi–Hubbard model consists of a lattice of coupled optical cavities, each containing a spin, which undergoes a phase transition from the Mott phase to the superfluid phase [34,35,36,37]. By partially breaking the exchange symmetry between the spins, the Dicke model can produce a quantum tricritical point [38]. A triangular structure formed by the Dicke model provides an opportunity to study the chiral properties and the geometric frustration [39,40,41]. Mapping the optomechanical problem of harmonically trapped atoms near a chiral waveguide to a generalized Rabi model reveals first-order quantum phase transitions with $Z_{3}$ symmetry breaking [42]. A quantum dot inside a single-mode cavity, described by the Rabi model, allows the distinguishing of topological phases when coupled to an additional Majorana nanowire [43]. Furthermore, the introduction of collective spin–spin interactions to the Dicke model stimulates the exploration of quantum criticality in systems combining matter-–matter and light-–matter interactions [44].

In this work, we investigate a generalized Dicke model including two spin ensembles within a single-mode cavity. The interaction between two spin ensembles can be ferromagnetic or antiferromagnetic, which themselves correspond to the well-known coupled-top model [21,45,46]. The interplay between spin–spin interaction and spin–boson interaction is expected to result in a more complex phase diagram and critical behavior.

The paper is structured as follows. In Section 2, we introduce the Hamiltonian of the generalized Dicke model. In Section 3, a phase diagram is constructed by employing the mean-field approach. Higher-order quantum effects beyond the mean-field contribution, such as the quantum fluctuation and the entanglement entropy, are given in Section 4. A brief summary is given in Section 5.

## 2. Hamiltonian

As a paradigmatic model to study the light–matter interaction, the Dicke model initially only introduced the interaction between a large ensemble of independent spins and a single-mode bosonic field [5,6,7]. In this paper, we introduce a generalized Dicke model as below, which includes two interacting spin ensembles coupled with a bosonic field:(1)$$\begin{array}{ccc} \hat{H} & = & {\hat{H}}_{S}+{\hat{H}}_{B}+{\hat{H}}_{I}, \end{array}$$(2)$$\begin{array}{ccc} {\hat{H}}_{S} & = & \Omega \left({\hat{J}}_{1,z}+{\hat{J}}_{2,z}\right)+\frac{\chi }{J}{\hat{J}}_{1,x}{\hat{J}}_{2,x}, \end{array}$$(3)$$\begin{array}{ccc} {\hat{H}}_{B} & = & \omega {\hat{b}}^{†}\hat{b}, \end{array}$$(4)$$\begin{array}{ccc} {\hat{H}}_{I} & = & \frac{\lambda }{\sqrt{J}}\left({\hat{J}}_{1,x}+{\hat{J}}_{2,x}\right)\left({\hat{b}}^{†}+\hat{b}\right). \end{array}$$
Here, the spin ensembles can be described by the angular momentum operator ${\hat{J}}_{i,s}=\sum_{n=1}^{N}{\hat{\sigma }}_{n,s}^{\left(i\right)}/2$, with s=x,y,z and i=1,2. The total number of spins in each ensemble is N=2J. ${\hat{H}}_{S}$ describes two interacting spin ensembles, where Ω is the frequency of the spin and χ is the spin–spin interacting strength. ${\hat{H}}_{S}$ is also known as the coupled-top model [21,45,46,47,48], which can be regarded as a generalization of the transverse-field Ising model. It can be realized by magnetic clusters coupled to superconducting loops of micro-SQUIDs [45]. For |Ω/χ|>1, two spin ensembles tend to align in parallel to the *z* axis, which leads to the paramagnetic phase. For |Ω/χ|<1, two spin ensembles prefer to align in parallel or anti-parallel along the *x* axis, depending on the sign of χ, which leads to ferromagnetic or anti-ferromagnetic phase. ${\hat{H}}_{B}$ describes the single-mode bosonic field with frequency ω. ${\hat{H}}_{I}$ represents the coupling between the spin ensembles and the bosonic field with spin–boson coupling strength λ.

The excitation number operators for spin ensembles and the bosonic field are defined as ${\hat{N}}_{S,i}={\hat{J}}_{i,z}+J$ and ${\hat{N}}_{B}={\hat{b}}^{†}\hat{b}$, respectively. In the absence of the spin–spin interaction (χ=0) and spin–boson coupling (λ=0), it is straightforward to confirm that the expectation values of the excitation number operators corresponding to the ground state are both zero, i.e., $\langle {\hat{N}}_{S,i}\rangle =\langle {\hat{N}}_{B}\rangle =0$. When both the spin–spin interaction and spin–boson coupling are weak enough ($\chi ,\lambda \ll \left\{\Omega ,\omega \right\}$), the rotating-wave approximation can be introduced. The spin–spin interaction and spin–boson coupling can be rewritten as
(5)$${\hat{J}}_{1,x}{\hat{J}}_{2,x}=\frac{1}{4}\left({\hat{J}}_{1,+}{\hat{J}}_{2,-}+{\hat{J}}_{1,-}{\hat{J}}_{2,+}\right)+\frac{1}{4}\left({\hat{J}}_{1,+}{\hat{J}}_{2,+}+{\hat{J}}_{1,-}{\hat{J}}_{2,-}\right),$$
(6)$$\begin{array}{ccc} \left({\hat{J}}_{1,x}+{\hat{J}}_{2,x}\right)\left({\hat{b}}^{†}+\hat{b}\right) & = & \frac{1}{2}\left[\left({\hat{J}}_{1,+}+{\hat{J}}_{2,+}\right)\hat{b}+\left({\hat{J}}_{1,-}+{\hat{J}}_{2,-}\right){\hat{b}}^{†}\right]\\ & + & \frac{1}{2}\left[\left({\hat{J}}_{1,+}+{\hat{J}}_{2,+}\right){\hat{b}}^{†}+\left({\hat{J}}_{1,-}+{\hat{J}}_{2,-}\right)\hat{b}\right], \end{array}$$
with ${\hat{J}}_{i,\pm }={\hat{J}}_{i,x}\pm i{\hat{J}}_{i,y}$. In general, the second term in Equations (Equation 5) and (Equation 6) is referred to as the counter-rotating term. The rotating-wave approximation involves disregarding this term, resulting in a Hamiltonian with U(1) symmetry and the conservation of the total number of excitations $\left({\hat{N}}_{S,1}+{\hat{N}}_{S,2}+{\hat{N}}_{B}\right)$ [2,5,12,29]. Unfortunately, the presence of the counter-rotating term disrupts the U(1) symmetry.

Nevertheless, the generalized Dicke model possesses a parity symmetry or $Z_{2}$ symmetry, given by $\left[\hat{H},\hat{\Pi }\right]=0$. The parity operator $\hat{\Pi }$ is defined as $\hat{\Pi }=\exp\left[i\pi \left({\hat{N}}_{S,1}+{\hat{N}}_{S,2}+{\hat{N}}_{B}\right)\right]$. The action of the parity transformation leads to $\hat{\Pi }{\hat{J}}_{i,x}{\hat{\Pi }}^{†}=-{\hat{J}}_{i,x}$, $\hat{\Pi }{\hat{J}}_{i,z}{\hat{\Pi }}^{†}={\hat{J}}_{i,z}$, $\hat{\Pi }\hat{b}{\hat{\Pi }}^{†}=-\hat{b}$, $\hat{\Pi }{\hat{b}}^{†}{\hat{\Pi }}^{†}=-{\hat{b}}^{†}$, while leaving the total Hamiltonian unaltered. In the symmetric phase, any eigenstate $|\psi \rangle$ of the Hamiltonian $\hat{H}$ must meet the condition $\hat{\Pi }|\psi \rangle =\Pi |\psi \rangle$, where Π=±1 signifies even and odd parities, respectively. It is straightforward to confirm that
(7a)$$\begin{array}{ccc} \langle \psi |{\hat{J}}_{i,x}|\psi \rangle & = & \langle \psi |{\hat{\Pi }}^{†}\hat{\Pi }{\hat{J}}_{i,x}{\hat{\Pi }}^{†}\hat{\Pi }|\psi \rangle =-\Pi^{2}\langle \psi |{\hat{J}}_{i,x}|\psi \rangle =-\langle \psi |{\hat{J}}_{i,x}|\psi \rangle , \end{array}$$
(7b)$$\begin{array}{ccc} \langle \psi |\hat{b}|\psi \rangle & = & \langle \psi |{\hat{\Pi }}^{†}\hat{\Pi }\hat{b}{\hat{\Pi }}^{†}\hat{\Pi }|\psi \rangle =-\Pi^{2}\langle \psi |\hat{b}|\psi \rangle =-\langle \psi |\hat{b}|\psi \rangle . \end{array}$$
Hence, in the symmetric phase, both the expectation values $\langle {\hat{J}}_{i,x}\rangle$ and $\langle \hat{b}\rangle$ are zero. However, when the parity symmetry is spontaneous breaking, the eigenstate $|\psi \rangle$ of the Hamiltonian will not be the eigenstate of $\hat{\Pi }$. Thus, $\langle {\hat{J}}_{i,x}\rangle$ and $\langle \hat{b}\rangle$ can be nonzero in the parity symmetry broken phase. This property renders them suitable as order parameters for determining the phase boundary [7,12,29,49]. Due to the competition between the spin–spin and spin–boson interactions, we expect that the generalized Dicke model exhibits fascinating phenomena, such as a richer phase diagram and novel critical behavior.

## 3. Mean-Field Approach

The mean-field approach is widely employed to investigate the Dicke model, coupled-top model, and their generalizations [8,9,50,51]. First, we discuss the mean-field representation of the generalized Dicke model, which provides a simple and intuitive physical picture despite the lack of correlations among different components. Prior to employing the mean-field approximation, we introduce the rotating operator ${\hat{R}}_{i}\left(\theta_{i}\right)$ and the displacement operator $\hat{D}\left(\alpha \right)$, defined as
(8a)$$\begin{array}{ccc} {\hat{R}}_{i}\left(\theta_{i}\right) & = & \exp\left(-i\theta_{i}{\hat{J}}_{i,y}\right)=\exp\left[\frac{\theta_{i}}{2}\left({\hat{J}}_{i,-}-{\hat{J}}_{i,+}\right)\right], \end{array}$$
(8b)$$\begin{array}{ccc} \hat{D}\left(\alpha \right) & = & \exp\left[\alpha \left({\hat{b}}^{†}-\hat{b}\right)\right]. \end{array}$$
In terms of ${\hat{R}}_{i}\left(\theta_{i}\right)$ and $\hat{D}\left(\alpha \right)$, the spin and bosonic coherent states, which are also known as the coherent states of Heisenberg–Weyl and SU(2) groups, can be expressed as
(9)$$\begin{array}{c} |\theta_{i}\rangle ={\hat{R}}_{i}\left(\theta_{i}\right)|J,-J\rangle ,\;|\alpha \rangle =\hat{D}\left(\alpha \right)|0\rangle , \end{array}$$
where $|J,-J\rangle$ represents the lowest Dicke state satisfying ${\hat{J}}_{i,z}|J,m\rangle =m|J,m\rangle$ with m=−J,−J+1,⋯,J, and $|0\rangle$ is the vacuum state of the bosonic field.

According to the mean-field theory, we construct a trial wave function formed by the tensor product of the spin and bosonic coherent states, namely,
(10)$$\begin{array}{c} |\psi_{\mathrm{MF}}\rangle =|\theta_{1}\rangle ⊗|\theta_{2}\rangle ⊗|\sqrt{N}\alpha \rangle . \end{array}$$
Then, we can calculate the average energy expectation value:(11)$$\begin{array}{ccc} E_{\mathrm{MF}}\left(\theta_{1},\theta_{2},\alpha \right) & = & \frac{1}{N}\langle \psi_{\mathrm{MF}}|\hat{H}|\psi_{\mathrm{MF}}\rangle \\ & = & -\frac{\Omega }{2}\left(\cos\theta_{1}+\cos\theta_{2}\right)+\frac{\chi }{2}\sin\theta_{1}\sin\theta_{2}+\omega \alpha^{2}-\sqrt{2}\lambda \alpha \left(\sin\theta_{1}+\sin\theta_{2}\right). \end{array}$$
Based on the variational principle, the unknown variational parameters $\theta_{1}$, $\theta_{2}$, and α can be determined by minimizing the average energy expectation value $E_{\mathrm{MF}}$, which leads to
(12a)$$\begin{array}{ccc} \frac{\partial E_{\mathrm{MF}}}{\partial \theta_{1}} & = & \frac{\Omega }{2}\sin\theta_{1}+\frac{\chi }{2}\sin\theta_{2}\cos\theta_{1}-\sqrt{2}\lambda \alpha \cos\theta_{1}=0, \end{array}$$
(12b)$$\begin{array}{ccc} \frac{\partial E_{\mathrm{MF}}}{\partial \theta_{2}} & = & \frac{\Omega }{2}\sin\theta_{2}+\frac{\chi }{2}\sin\theta_{1}\cos\theta_{2}-\sqrt{2}\lambda \alpha \cos\theta_{2}=0, \end{array}$$
(12c)$$\begin{array}{ccc} \frac{\partial E_{\mathrm{MF}}}{\partial \alpha } & = & 2\left(\omega \alpha -\frac{\lambda }{\sqrt{2}}\left(\sin\theta_{1}+\sin\theta_{2}\right)\right)=0. \end{array}$$
Upon determination of $\theta_{1}$, $\theta_{2}$, and α, the ground-state energy $E_{\mathrm{MF}}$ and wave function $|\psi_{\mathrm{MF}}\rangle$ are achieved. Subsequently, the order parameters $\langle {\hat{J}}_{i,x}\rangle$ and $\langle \hat{b}\rangle$ can be calculated accordingly, with
(13)$$\begin{array}{ccc} \langle {\hat{J}}_{i,x}\rangle & = & \langle \psi_{\mathrm{MF}}|{\hat{J}}_{i,x}|\psi_{\mathrm{MF}}\rangle =-J\sin\theta_{i}, \end{array}$$
(14)$$\begin{array}{ccc} \langle \hat{b}\rangle & = & \langle \psi_{\mathrm{MF}}|\hat{b}|\psi_{\mathrm{MF}}\rangle =\sqrt{N}\alpha , \end{array}$$
from which we can find out whether the quantum phase transition exists. The order of the quantum phase transition can be determined by the derivative of the ground-state energy $E_{\mathrm{MF}}$ with respective to system parameters [12,29]. A first-order quantum phase transition is indicated by a discontinuous first derivative, $dE_{\mathrm{MF}}/d\chi$, while a second-order quantum phase transition is marked by a discontinuous second derivative, $d^{2}E_{\mathrm{MF}}/d\chi^{2}$. Furthermore, the excitation numbers $\langle {\hat{N}}_{S,i}\rangle$ and $\langle {\hat{N}}_{B}\rangle$ also offer some insights in different phases, given by
(15)$$\begin{array}{ccc} \langle {\hat{N}}_{s,i}\rangle & = & \langle \psi_{\mathrm{MF}}|{\hat{J}}_{i,z}|\psi_{\mathrm{MF}}\rangle +J=J\left(1-\cos\theta_{i}\right), \end{array}$$
(16)$$\begin{array}{ccc} \langle {\hat{N}}_{B}\rangle & = & \langle \psi_{\mathrm{MF}}|{\hat{N}}_{B}|\psi_{\mathrm{MF}}\rangle =N\alpha^{2}. \end{array}$$

Based on the analysis above, Figure 1 displays the phase diagram of the generalized Dicke model, revealing three distinct phases. The dashed line is represented by $\chi =\frac{4\lambda^{2}-\Omega \omega }{\omega }$. This line represents the second-order quantum phase transition that distinguishes Phase I from Phase II, as indicated by the continuous $dE_{\mathrm{MF}}/d\chi$ in Figure 1a and discontinuous $d^{2}E_{\mathrm{MF}}/d\chi^{2}$ in Figure 1b. The dotted line is expressed as χ=Ω. It signifies the second-order quantum phase transition that separates Phase I from Phase III, characterized by the discontinuous $d^{2}E_{\mathrm{MF}}/d\chi^{2}$ in Figure 1b. The solid line is expressed as $\chi =\frac{2\lambda^{2}}{\omega }$. It corresponds to the first-order quantum phase transition that separates Phase II from Phase III, as reflected in the discontinuous $dE_{\mathrm{MF}}/d\chi$ in Figure 1a.

Phase I is present only if both the spin–spin interaction strength |χ| and the spin–boson coupling strength λ are small enough. Solving Equation (12) results in $\theta_{1}=\theta_{2}=0$ and α=0. The ground state is nondegenerate, with energy $E_{\mathrm{MF}}=-\Omega$. Moreover, the parity symmetry is unbroken, as indicated by $\langle {\hat{J}}_{i,x}\rangle =0$ in Figure 1c, and $\langle \hat{b}\rangle =0$ in Figure 1d. The spin ensembles tend to align in parallel to the *z* axis due to $\langle {\hat{J}}_{i,z}\rangle /J=-1$, which corresponds to the paramagnetic phase in the coupled-top model. There are no macroscopic excitations in the bosonic field due to $\langle {\hat{N}}_{B}\rangle =0$, which is consistent with what happens in the normal phase of the original Dicke model. To sum up, Phase I is referred to as the paramagnetic–normal phase.

Phase II is located in the lower-right corner of Figure 1, which corresponds to the ferromagnetic–superradiant phase. The energy minimum occurs at $\theta_{1}=\theta_{2}=\theta =\pm \arccos\left(\frac{\Omega \omega }{4\lambda^{2}-\chi \omega }\right)$ and $\alpha =\frac{\sqrt{2}\lambda \sin\theta }{\omega }$, which corresponds to a twofold degenerate ground state with energy
(17)$$\begin{array}{ccc} E_{\mathrm{MF}} & = & -\frac{\Omega }{2}\left(\frac{\Omega \omega }{4\lambda^{2}-\chi \omega }+\frac{4\lambda^{2}-\chi \omega }{\Omega \omega }\right). \end{array}$$
From Equations (Equation 13) and (Equation 14), it is easy to confirm that $\langle J_{1,x}\rangle =\langle J_{2,x}\rangle \neq 0$ and $\langle \hat{b}\rangle \neq 0$, which is the signature of parity symmetry breaking. Phase II is termed the ferromagnetic–superradiant phase due to the following reasons:When the spin–boson coupling strength λ is held constant, the critical spin–spin interaction strength that distinguishes Phase I from Phase II is given by $\chi =\frac{4\lambda^{2}-\Omega \omega }{\omega }$. In the case of λ=0, the generalized Dicke model is reduced to the coupled-top model, with a critical point χ=−Ω. In the coupled-top model, it is well-known that a strong ferromagnetic spin–spin interaction (χ<−Ω) is required to observe the ferromagnetic phase, where two spin ensembles prefer to align in parallel along the *x* axis. From this perspective, Phase II corresponds to the ferromagnetic phase, as $\langle J_{1,x}\rangle =\langle J_{1,x}\rangle \neq 0$. The spin–boson coupling promotes the formation of the ferromagnetic phase by significantly reducing the ferromagnetic spin–spin interaction strength |χ| required to induce the phase transition. The ferromagnetic phase persists even in the presence of antiferromagnetic spin–spin interaction (χ>0), provided that λ is sufficiently large;When the spin–spin interaction strength χ is held constant, the critical spin–boson coupling strength that distinguishes Phase I from Phase II is given by $\lambda =\frac{\sqrt{\left(\Omega +\chi \right)\omega }}{2}$. In the case of χ=0, the generalized Dicke model is reduced to the original Dicke model, with a critical point $\lambda =\sqrt{\Omega \omega }/2$. In the original Dicke model, it is well-known that a strong spin–boson coupling ($\lambda >\sqrt{\Omega \omega }/2$) is required to observe the superradiant phase, where macroscopic excitations of the bosonic field emerge. From this perspective, Phase II corresponds to the superradiant phase, as indicated by $\langle {\hat{N}}_{B}\rangle >0$. The antiferromagnetic spin–spin interaction (χ>0) hinders the formation of the superradiance, whereas the ferromagnetic spin–spin interaction (χ<0) promotes the formation of the superradiant phase.

Phase III is situated in the upper-left corner of Figure 1. The energy minimum is reached when $\theta_{1(2)}=-\theta_{2(1)}=\pm \arccos\left(\frac{\Omega }{\chi }\right)$ and α=0, leading to twofold degenerate ground states, with energy $E_{\mathrm{MF}}=-\frac{\Omega }{2}\left(\frac{\Omega }{\chi }+\frac{\chi }{\Omega }\right)$. Varying the spin–boson coupling λ does not influence the critical spin–spin interaction strength χ that separates Phase I and III. From Equations (Equation 13) and (Equation 14), we find that $\langle {\hat{J}}_{1,x}\rangle =-\langle {\hat{J}}_{2,x}\rangle \neq 0$, which indicates the breaking of the parity symmetry. Two spin ensembles prefer to align anti-parallelly along the *x* axis, which is consistent with the behavior observed in the antiferromagnetic phase of the coupled-top model. It should be noted that the original Dicke model enters the superradiant phase with macroscopic excitations in the bosonic field as long as the parity symmetry is breaking [11]. However, as indicated in Figure 1d, Phase III exhibits no macroscopic excitations in the bosonic field, regardless of the strength of the spin–boson coupling. Therefore, Phase III corresponds to the normal phase in the original Dicke model. Overall, Phase III is referred to as a ferromagnetic–normal phase.

## 4. Beyond the Mean-Field Approach

As illustrated in the previous section, the mean-field approach elucidates the behaviors of the order parameters, from which we can capture the phase diagram. Nonetheless, higher-order contributions, such as the quantum fluctuation, correlation and entanglement [49,52,53], are obscured, highlighting the urgent demand for methods extending beyond the mean-field approach. A commonly employed technique for separating mean-field and higher-order contributions involves a unitary transformation $\hat{U}={\hat{R}}_{1}\left(\theta_{1}\right){\hat{R}}_{2}\left(\theta_{2}\right)\hat{D}\left(\sqrt{N}\alpha \right)$ [41,54,55,56], namely, $\hat{\bar{H}}={\hat{U}}^{†}\hat{H}\hat{U}$, with $\theta_{1}$, $\theta_{2}$ and α determined through the mean-field approach in Section 3. Since the rotating operator ${\hat{R}}_{i}\left(\theta_{i}\right)$ and the displacement operator $\hat{D}\left(\sqrt{N}\alpha \right)$ satisfy the following properties:
(18a)$$\begin{array}{cc} {\hat{R}}_{i}^{†}\left(\theta_{i}\right){\hat{J}}_{i,x}{\hat{R}}_{i}\left(\theta_{i}\right)=\cos\theta_{i}{\hat{J}}_{i,x}+\sin\theta_{i}{\hat{J}}_{i,z}, & \;{\hat{D}}^{†}\left(\sqrt{N}\alpha \right)\hat{b}\hat{D}\left(\sqrt{N}\alpha \right)=\hat{b}+\sqrt{N}\alpha , \end{array}$$
(18b)$$\begin{array}{cc} {\hat{R}}_{i}^{†}\left(\theta_{i}\right){\hat{J}}_{i,z}{\hat{R}}_{i}\left(\theta_{i}\right)=\cos\theta_{i}{\hat{J}}_{i,z}-\sin\theta_{i}{\hat{J}}_{i,x}, & \;{\hat{D}}^{†}\left(\sqrt{N}\alpha \right){\hat{b}}^{†}\hat{D}\left(\sqrt{N}\alpha \right)={\hat{b}}^{†}+\sqrt{N}\alpha , \end{array}$$
one can easily achieve the transformed Hamiltonian $\hat{\bar{H}}$. Subsequently, the Holstein–Primakoff transformation [10,11,57] is introduced to map the angular momentum operators into bosonic creation and annihilation operators as
(19)$$\begin{array}{c} {\hat{J}}_{i,z}={\hat{a}}_{i}^{†}{\hat{a}}_{i}-J,\;{\hat{J}}_{i,+}={\hat{a}}_{i}^{†}\sqrt{N-{\hat{a}}_{i}^{†}{\hat{a}}_{i}},\;{\hat{J}}_{i,-}=\sqrt{N-{\hat{a}}_{i}^{†}{\hat{a}}_{i}}{\hat{a}}_{i}. \end{array}$$
In the thermodynamic limit (N→+∞), it is anticipated that $N\gg \langle {\hat{a}}_{i}^{†}{\hat{a}}_{i}\rangle$, and the Holstein–Primakoff transformation can be simplified as
(20)$$\begin{array}{c} {\hat{J}}_{i,z}={\hat{a}}_{i}^{†}{\hat{a}}_{i}-J,\;{\hat{J}}_{i,+}\approx \sqrt{N}{\hat{a}}_{i}^{†},\;{\hat{J}}_{i,-}\approx \sqrt{N}{\hat{a}}_{i}, \end{array}$$
After the Holstein-Primakoff transformation, we can write $\hat{\bar{H}}$ as a series expansion in powers of 1/N as follows,
(21)$$\begin{array}{ccc} \hat{\bar{H}} & \approx & {\left(\frac{1}{N}\right)}^{-1}E_{\mathrm{MF}}\left(\theta_{1},\theta_{2},\alpha \right)+{\left(\frac{1}{N}\right)}^{-1/2}{\hat{\bar{H}}}_{1}+{\left(\frac{1}{N}\right)}^{0}{\hat{\bar{H}}}_{2}, \end{array}$$
with
(22)$$\begin{array}{ccc} {\hat{\bar{H}}}_{1} & = & -\left(\frac{\Omega }{2}\sin\theta_{1}+\frac{\chi }{2}\sin\theta_{2}\cos\theta_{1}-\sqrt{2}\lambda \alpha \cos\theta_{1}\right)\left({\hat{a}}_{1}^{†}+{\hat{a}}_{1}\right)\\ & & -\left(\frac{\Omega }{2}\sin\theta_{2}+\frac{\chi }{2}\sin\theta_{1}\cos\theta_{2}-\sqrt{2}\lambda \alpha \cos\theta_{2}\right)\left({\hat{a}}_{2}^{†}+{\hat{a}}_{2}\right)\\ & & +\left(\omega \alpha -\frac{\lambda }{\sqrt{2}}\left(\sin\theta_{1}+\sin\theta_{2}\right)\right)\left({\hat{b}}^{†}+\hat{b}\right), \end{array}$$
(23)$$\begin{array}{ccc} {\hat{\bar{H}}}_{2} & = & \frac{\chi }{2}\cos\theta_{1}\cos\theta_{2}\left({\hat{a}}_{1}^{†}+{\hat{a}}_{1}\right)\left({\hat{a}}_{2}^{†}+{\hat{a}}_{2}\right)\\ & & +\frac{\lambda }{\sqrt{2}}\left[\cos\theta_{1}\left({\hat{a}}_{1}^{†}+{\hat{a}}_{1}\right)+\cos\theta_{2}\left({\hat{a}}_{2}^{†}+{\hat{a}}_{2}\right)\right]\left({\hat{b}}^{†}+\hat{b}\right)\\ & & +\left(\Omega \cos\theta_{1}-\chi \sin\theta_{1}\sin\theta_{2}+2\sqrt{2}\lambda \alpha \sin\theta_{1}\right){\hat{a}}_{1}^{†}{\hat{a}}_{1}\\ & & +\left(\Omega \cos\theta_{2}-\chi \sin\theta_{1}\sin\theta_{2}+2\sqrt{2}\lambda \alpha \sin\theta_{2}\right){\hat{a}}_{2}^{†}{\hat{a}}_{2}\\ & & +\omega {\hat{b}}^{†}\hat{b}. \end{array}$$
where we have ignored the higher-order terms proportional to ${\left(1/N\right)}^{l}$ with l>0. The first term in Equation (Equation 21) is a constant, representing the ground-state energy (Equation 17) derived from the mean-field approach. $\theta_{1}$, $\theta_{2}$ and α obtained from Equation (12) lead to ${\hat{\bar{H}}}_{1}=0$, which further simplifies the low-energy effective Hamiltonian. Finally, we only need to deal with the quadratic Hamiltonian ${\hat{\bar{H}}}_{2}$.

By introducing ${\hat{x}}_{i}=\left({\hat{a}}_{i}^{†}+{\hat{a}}_{i}\right)/\sqrt{2}$, ${\hat{p}}_{i}=i\left({\hat{a}}_{i}^{†}-{\hat{a}}_{i}\right)/\sqrt{2}$, for i=1,2, and ${\hat{x}}_{3}=\left({\hat{b}}^{†}+\hat{b}\right)/\sqrt{2}$, ${\hat{p}}_{3}=i\left({\hat{b}}^{†}-\hat{b}\right)/\sqrt{2}$, the quadratic Hamiltonian ${\hat{\bar{H}}}_{2}$ can be rewritten as
(24)$$\begin{array}{ccc} {\hat{\bar{H}}}_{2} & = & \sum_{i=1}^{3}\frac{\epsilon_{i}}{2}\left({\hat{x}}_{i}^{2}+{\hat{p}}_{i}^{2}-1\right)+\tau_{i,i+1}{\hat{x}}_{i}{\hat{x}}_{i+1}, \end{array}$$
with
(25a)$$\begin{array}{ccc} \epsilon_{1} & = & \Omega \cos\theta_{1}-\chi \sin\theta_{1}\sin\theta_{2}+2\sqrt{2}\lambda \alpha \sin\theta_{1}, \end{array}$$
(25b)$$\begin{array}{ccc} \epsilon_{2} & = & \Omega \cos\theta_{2}-\chi \sin\theta_{1}\sin\theta_{2}+2\sqrt{2}\lambda \alpha \sin\theta_{2}, \end{array}$$
(25c)$$\begin{array}{ccc} \epsilon_{3} & = & \omega , \end{array}$$
(25d)$$\begin{array}{ccc} \tau_{1,2} & = & \chi \cos\theta_{1}\cos\theta_{2}=\tau_{2,1}, \end{array}$$
(25e)$$\begin{array}{ccc} \tau_{1,3} & = & \sqrt{2}\lambda \cos\theta_{1}=\tau_{3,1}, \end{array}$$
(25f)$$\begin{array}{ccc} \tau_{2,3} & = & \sqrt{2}\lambda \cos\theta_{2}=\tau_{3,2}. \end{array}$$
Clearly, this corresponds to coupled harmonic oscillators. As demonstrated in Appendix A, this Hamiltonian can be solved exactly using the symplectic transformation [58,59], which decouples the coupled harmonic oscillators into the following form:(26)$$\begin{array}{c} {\hat{\bar{H}}}_{2}=\sum_{i=1}^{3}\frac{\Delta_{i}}{2}\left({\hat{x}}_{i}^{\prime 2}+{\hat{p}}_{i}^{\prime 2}\right)-\sum_{i=1}^{3}\frac{\epsilon_{i}}{2}. \end{array}$$
$\Delta_{i}\geq 0$ corresponds to the excitation energy, as shown in Figure 2a. Without loss of generality, we select the parameters Ω/ω=1 and λ/ω=0.3, while allowing χ/ω to range from −2 to 2. This range encompasses all three phases, as depicted in Figure 1. The critical point that separates Phase I from Phase II is situated at $\chi_{c-}/\omega =-0.64$, whereas the critical point separating Phase I and Phase III is found at $\chi_{c+}/\omega =1$. The lowest excitation energy, namely, $\Delta_{\min}=\min\left(\Delta_{1},\Delta_{2},\Delta_{3}\right)$, represents the energy gap between the ground state and the first excited state. Generally, the quantum phase transition occurs concurrently with the closing of the energy gap ($\Delta_{\min}\to 0$), which is consistent with our results in Figure 2a. As illustrated in Figure 2b,c, the critical behavior associated with the excitation energy is given by $\Delta_{\min}\propto {\left|\chi -\chi_{c\pm }\right|}^{1/2}$ near the critical point $\chi_{c\pm }$, which is in accordance with that in the original Dicke model [10,57] and the coupled-top model [56].

In stark contrast to its classical counterpart driven by thermal fluctuations, the quantum phase transition takes place at zero temperature due to the quantum fluctuations [15]. The quantum fluctuations in $x_{i}$ and $p_{i}$ quadrature are expressed as
(27)$$\begin{array}{c} {\left(\Delta x_{i}\right)}^{2}=\langle {\hat{x}}_{i}^{2}\rangle -{\langle {\hat{x}}_{i}\rangle }^{2},\;{\left(\Delta p_{i}\right)}^{2}=\langle {\hat{p}}_{i}^{2}\rangle -{\langle {\hat{p}}_{i}\rangle }^{2}. \end{array}$$
Figure 2d,g depicts the behaviors of the quantum fluctuations in $x_{i}$ and $p_{i}$ quadrature, respectively. Since both i=1,2 correspond to the spin ensembles, we only show one of them for clarity. Far away from the critical point $\chi_{c\pm }$, both ${\left(\Delta x_{i}\right)}^{2}$ and ${\left(\Delta p_{i}\right)}^{2}$ tend to approach 1/2, which can be well captured by the coherent state in the mean-field approach. Near the critical point $\chi_{c\pm }$, ${\left(\Delta p_{i}\right)}^{2}$ becomes less than 1/2, which indicates a strong squeezing effect [2]. All three quantum fluctuations in $x_{i}$ quadrature tend to exponentially diverge with ${\left(\Delta x_{i}\right)}^{2}\propto {\left|\chi -\chi_{c-}\right|}^{-1/2}$ near the critical point $\chi_{c-}$, as shown in Figure 2e. Similar phenomena can be found for $x_{1}$ and $x_{2}$ quadrature near the critical point $\chi_{c+}$, both of which correspond to the spin components. Nevertheless, Figure 2f indicates that $x_{3}$ quadrature, associated with the bosonic field, does not exhibit an exponential divergent fluctuation near $\chi_{c+}$. As discussed in Section 3, $\chi_{c+}$ separates Phase I and Phase III, which correspond to paramagnetic–normal phase and antiferromagnetic–normal phase, respectively. The quantum phase transition is dominated by the antiferromagnetic spin–spin interactions, leading to substantial fluctuations in the spin components, as opposed to the bosonic field.

The entanglement entropy, also known as the von Neumann entropy, is proposed to quantify the entanglement between different components of the quantum system [60,61]. It is directly associated with the Heisenberg’s uncertainty relation for the quadratic Hamiltonian of interacting bosonic systems [52,61]. In terms of $\Delta x_{i}$ and $\Delta p_{i}$, the entanglement entropy $S_{i}$ can be written as
(28)$$\begin{array}{c} S_{i}=\left(\Delta x_{i}\Delta p_{i}+\frac{1}{2}\right)\log\left(\Delta x_{i}\Delta p_{i}+\frac{1}{2}\right)-\left(\Delta x_{i}\Delta p_{i}-\frac{1}{2}\right)\log\left(\Delta x_{i}\Delta p_{i}-\frac{1}{2}\right), \end{array}$$
which describes the entanglement between the *i*th component and the others. Recently, there has been growing interest in investigating quantum phase transitions from the perspective of entanglement [38,45,49,52,62]. As depicted in Figure 2h, the entanglement entropies $S_{1}$ and $S_{3}$ exhibit divergences near the quantum critical point $\chi_{c-}$, which indicate strong entanglement among two spin ensembles and one bosonic field. However, $S_{3}$ is negligible compared to $S_{1}$ near $\chi_{c+}$, which indicates that the bosonic field is almost independent from spin ensembles. The finite $S_{3}$ near $\chi_{c+}$ has a similar origin to the finite quantum fluctuation ${\left(\Delta x_{3}\right)}^{2}$. Both Phase I and Phase III, separated by $\chi_{c+}$, exhibit no macroscopic excitations in the bosonic field. The quantum phase transition is primarily driven by the strong antiferromagnetic spin–spin interaction, leading to significant entanglement between two spin ensembles, as indicated by the divergence in $S_{1}$. In contrast, the spin–boson coupling has a negligible effect, resulting in weak entanglement between the bosonic field and the two spin ensembles.

## 5. Conclusions

The Dicke model serves as a paradigmatic model to study the light–matter interaction, where the bosonic field represents light and the spin ensemble represents matter. It undergoes a quantum phase transition from the normal phase to the superradiant phase for sufficient strong spin–boson coupling. The coupled-top model describe two interacting spin ensembles, which can be regarded as an example of matter–matter interaction. It exhibits paramagnetic phase, ferromagnetic phase and antiferromagnetic phase, depending on the spin–spin interaction strength. In this work, we proposed a generalized Dicke model that combines the light–matter interaction in the Dicke model with the matter–matter interaction in the coupled-top model. This is achieved by introducing two interacting spin ensembles coupled with a bosonic field.

Due to the competition between the spin–spin interaction and the spin–boson coupling, the generalized Dicke model admits a diverse phase diagram, which consists of three phases: paramagnetic–normal phase, ferromagnetic–superradiant phase and antiferromagnetic–normal phase. The paramagnetic–normal phase is present only if both the spin–spin interaction and the spin–boson coupling are sufficiently weak. In this phase, two spin ensembles tend to align in parallel along the *z* axis, while the bosonic field prohibits macroscopic excitation. In the ferromagnetic–superradiant phase, two spin ensembles prefer to align in parallel along the *x* axis, while the macroscopic excitation emerges in the bosonic field. Interestingly, the spin–boson coupling strength required to stimulate the macroscopic excitation is significantly suppressed in the presence of the ferromagnetic spin–spin interaction. In the antiferromagnetic–normal phase, two spin ensembles prefer to align anti-parallelly along the *x* axis. No macroscopic excitation in the bosonic field emerges, regardless of the strength of the spin–boson coupling.

The boundary of the phase diagram can be distinguished by the mean-field approach. Nevertheless, it falls short in offering deeper insights into the higher-order quantum effects, such as the excitation energy, quantum fluctuation and entanglement entropy. These can be achieved through the utilization of the Holstein–Primakoff transformation and the symplectic transformation, which transform the generalized Dicke into three decoupled harmonic oscillators in the thermodynamic limit. The excitation energy approaches zero in the vicinity of the critical point. The closing of the energy gap between the ground and the first excited states coincides with the divergence of the quantum fluctuation in certain quadrature and the entanglement entropy. Our generalizations to the Dicke model further improve its flexibility and open up new opportunities to investigate the competition between light–matter and matter–matter interactions.

It is worth mentioning that approaches both within and beyond the mean-field theory are performed in the thermodynamic limit in this work. Recently, the finite-component system has drawn renewed attention. These systems not only offer enhanced experimental accessibility but also yield valuable insights into quantum phase transitions by revealing finite-size scaling behavior near critical points [13,14,62,63]. Moreover, quantum phase transitions and spontaneous symmetry breaking can even exist in the finite-component system, such as the Rabi model and its generalizations [18,27,32,49,50,64]. The finite-size effects in the generalized Dicke model deserve further consideration, which are left to future research.

## Figures and Tables

**Figure 1 entropy-25-01492-f001:**
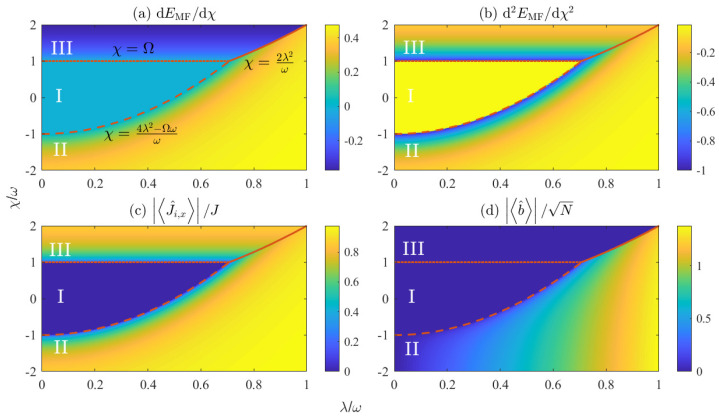
(**a**) $dE_{\mathrm{MF}}/d\chi$, (**b**) $d^{2}E_{\mathrm{MF}}/d\chi^{2}$, (**c**) $\left|\langle {\hat{J}}_{i,x}\rangle \right|/J$, and (**d**) $\left|\langle \hat{b}\rangle \right|/\sqrt{N}$ as a function of the spin–spin coupling strength χ and the spin–boson coupling strength λ at Ω/ω=1. The dashed line denotes the phase boundary separating Phase I and II, which can be expressed as $\chi =\frac{4\lambda^{2}-\Omega \omega }{\omega }$. The dotted line denotes the phase boundary separating Phase I and III, which can be expressed as χ=Ω. The solid line denotes the phase boundary separating Phase II and III, which can be expressed as $\chi =\frac{2\lambda^{2}}{\omega }$.

**Figure 2 entropy-25-01492-f002:**
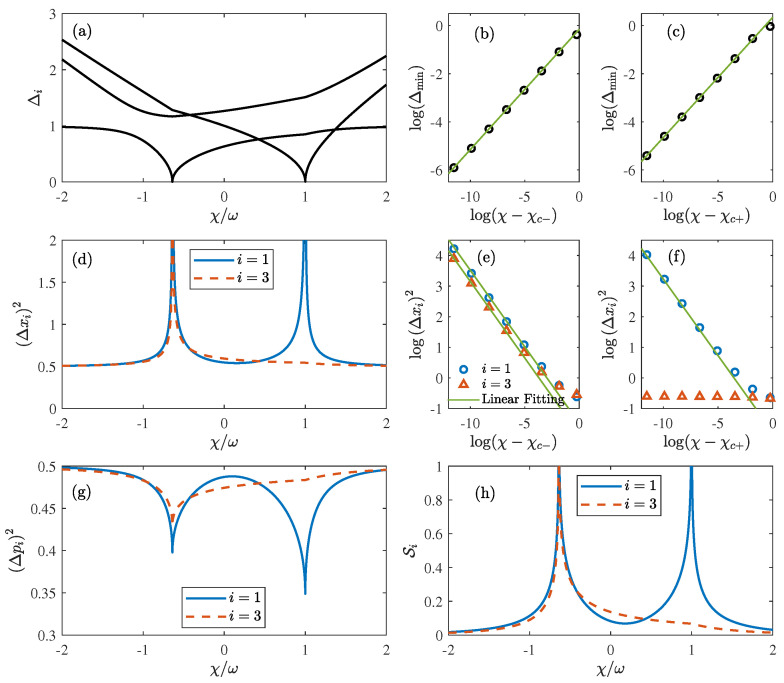
(**a**) The excitation energy $\Delta_{i}$, (**d**) the quantum fluctuation ${\left(\Delta x_{1}\right)}^{2}$ (blue solid line) and ${\left(\Delta x_{3}\right)}^{2}$ (red dashed line), (**g**) the quantum fluctuation ${\left(\Delta p_{1}\right)}^{2}$ (blue solid line) and ${\left(\Delta p_{3}\right)}^{2}$ (red dashed line) and (**h**) the entanglement entropy $S_{1}$ (blue solid line) and $S_{3}$ (red dashed line) as a function of χ for Ω/ω=1, λ/ω=0.3. The critical points are located at $\chi_{c-}/\omega =-0.64$ and $\chi_{c+}/\omega =1$. (**b**,**c**) show the critical behaviors associated with the lowest excitation energy $\Delta_{\min}$ near $\chi_{c-}$ and $\chi_{c+}$, respectively. The green line corresponds to a linear fitted curve with a slope of $\frac{1}{2}$. (**e**,**f**) show the critical behaviors associated with the quantum fluctuation ${\left(\Delta x_{1}\right)}^{2}$ (blue circle) and ${\left(\Delta x_{3}\right)}^{2}$ (red triangle) near $\chi_{c-}$ and $\chi_{c+}$, respectively. The green line corresponds to a linear fitted curve with a slope of $-\frac{1}{2}$.

## Data Availability

The data that support the findings of this study are available from the corresponding author upon reasonable request.

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
