# Peer review of "Quantum Phase Transitions in a Generalized Dicke Model"

_entropy, 2023, doi:10.3390/e25111492_

Round 1

Reviewer 1 Report

Comments and Suggestions for Authors

In this work, the Rabi model of N isotopic qubits coupled to a monochromatic bosonic mode is modified by considering two different ensembles of N isotropic qubits coupled to the same monochromatic mode. The specific choice of the interactions among the two isotropic spin ensembles was an all-to-all exchange interaction. This article focusses on determining the phase diagram of the model and that the phase transitions possess the properties of the zero temperature quantum phase transitions, and comparisons of the mean-field and beyond mean-field properties to those known for the component models.

The paper is focussed on determining the quantum phase transitions of the new quantum optical model in both the mean-field and some aspects of the first order quantum effects around the phase transition points. The well-defined mean-field treatment reveals a more complex phase diagram then present in the Dicke model due to the existence of the all- to-all interaction between the spins. Additionally the first order quantum correction to the mean field Hamiltonian revealed using the Holstein-Primakoff transformation are shown to be consistent with the mean field results and defines a true quantum phase transition.

The paper could be published after considering the following comments.

1. In the first paragraph of the introduction it may be worth adding a sentence pointing to  the Review/Introduction to the physics of the of the Dicke model by Kirton et al Adv. Quantum Tech. 2, 1800043 (2019).

2. When commenting on the conditions for the preservation of total excitation around Line 82 it should be briefly specified what the rotating-wave approximation does to the interaction term of the Hamiltonian.

3. When commenting on the suitability of the parameters ⟨Jˆ ⟩ and ⟨ˆb⟩ on Lines 88-90 for determining the presence of the phase boundary a reference/discussion should be supplied to highlight the importance of the parameters and phase boundary.

4. When describing the relevant phases in Section 3 it may be worth placing more emphasis on where the nature of the phase transitions (expectation values of parameters and boundary conditions) differ from the Dicke model.

5. Figures should be provided to demonstrate that the critical exponents mentioned in Section 4 are a good fit to the obtained energy separations and variance of the position quadrature.

6. The properties of S1 around χc+ are discussed on line 236. The author is asked whether a comment can be added to link this result to the quadrature x3 in the same region and the mean field bosonic excitation accords the boundary between Phases I and III.

7. For completeness within the Appendix, the author additionally provides the form of the symplectic transformation matrix S as it is a key part of the outlined derivation.

Reviewer 2 Report

Comments and Suggestions for Authors

The manuscript explores an interesting interplay between matter-matter and light-matter interactions in a generalized Dicke model, that is comprised of two Dicke models collectively coupled to each other. The findings offer insights into the quantum phase transitions and critical behaviors of the model, with a focus on the rich phase diagram it exhibits. Before I make my final conclusion I want a few issues/questions to be addressed: 

1) In your conclusion, you mention the limitations of the mean-field approach in providing deeper insights into higher-order quantum effects.  Did you compare the mean-field approach to the exact diagonalization and if yes, what size of the system is enough for the mean-field to give accurate prediction?

2) Could you clarify if there are any systems or setups where one may have such a collective spin-spin coupling? I believe this should be commented on in the manuscript

3) To strengthen the manuscript further, I recommend citing [https://doi.org/10.1103/PhysRevLett.125.263606] to draw parallels with existing research on quantum phase transitions in Dicke-like models. Additionally, incorporating [https://doi.org/10.1103/PhysRevA.106.023702] could help readers understand the potential relevance of cavity setups in distinguishing topological phases.

Overall, this manuscript could be potentially interesting to the community working on light-matter interaction. Addressing these questions and incorporating the suggested references could enhance the manuscript's impact and clarity.

Comments on the Quality of English Language

recommend running the manuscript through software that would detect and correct multiple misprints, presented in the text

Reviewer 3 Report

Comments and Suggestions for Authors

The authors study investigate a generalized Dicke model by introducing two interacting spin ensembles
coupled with a single-mode bosonic field. Before I give a final conclusion, they should reply to the following comments.

The sentence "The Dicke model provides
a fundamental framework for exploring collective quantum behavior and the intricate
interplay between quantum systems and electromagnetic fields.'' should be cited with recent related references, for example,
1.A similarity of quantum phase transition and quench dynamics in the Dicke model beyond the thermodynamic limit. EPJ Quantum Technology (2020) 7, 1 and so on.

For the sentences ``...corresponding to the second-order quantum phase transition that separates Phase I from Phase II...'' and ``corresponding to the first-order quantum phase transition that separates Phase II from Phase III.'' Why?They should give a quantitative explanation, not just a result. i.e., how to prove that the quantum phase transition is first order or second order?

The thermodynamic limit can not be realized in fact. They do not give the discussions of the effect of finite N, i.e., how N affects the quantum phase transition.

Comments on the Quality of English Language

Minor editing of English language required

Round 2

Reviewer 1 Report

Comments and Suggestions for Authors

The author has successfully addressed comments and I recommend the publication of this article in Entropy in the present form.

Reviewer 2 Report

Comments and Suggestions for Authors

I am satisfied with the authors' response 

Comments on the Quality of English Language

I am not a proofreading editor, but to me the English is fine

Reviewer 3 Report

Comments and Suggestions for Authors

I agree to accept this version of manuscript.